# Optical Nonlinearity of ZrS_2_ and Applications in Fiber Laser

**DOI:** 10.3390/nano9030315

**Published:** 2019-02-27

**Authors:** Lu Li, Ruidong Lv, Jiang Wang, Zhendong Chen, Huizhong Wang, Sicong Liu, Wei Ren, Wenjun Liu, Yonggang Wang

**Affiliations:** 1School of Science, Xi’an University of Posts and Telecommunications, Xi’an 710121, China; liluyoudian@163.com; 2School of Physics and information Technology, Shanxi Normal University, Xi’an 710119, China; ruidonglv@126.com (R.L.); j_wang@snnu.edu.cn (J.W.); 13071095760@163.com (Z.C.); SNUwhz@163.com (H.W.); liusicong8@163.com (S.L.); chinawygxjw@snnu.edu.cn (Y.W.); 3State Key Laboratory of Information Photonics and Optical Communications, School of Science, Beijing University of Posts and Telecommunications, Beijing 100876, China

**Keywords:** ZrS_2_, saturable absorbers, transition metal dichalcogenides

## Abstract

Group VIB transition metal dichalcogenides (TMDs) have been successfully demonstrated as saturable absorbers (SAs) for pulsed fiber lasers. For the group comprising IVB TMDs, applications in this field remain unexplored. In this work, ZrS_2_-based SA is prepared by depositing a ZrS_2_ nanostructured film onto the side surface of a D-shaped fiber. The nonlinear optical properties of the prepared SA are investigated, which had a modulation depth of 3.3% and a saturable intensity of 13.26 MW/cm^2^. In a pump power range of 144–479 mW, the Er-doped fiber (EDF) laser with ZrS_2_ can operate in the dual-wavelength Q-switching state. The pulse duration declined from 10.0 μs down to 2.3 μs. The single pulse energy reached 53.0 nJ. The usage of ZrS_2_ as a SA for pulse generation in fiber lasers is presented for the first time. Compared to the experimental results of dual-wavelength Q-switched fiber lasers with two-dimensional (2D) materials, our laser performance was better. Our work indicates that the group comprising IVB TMD ZrS_2_ has bright prospects for nonlinear optical applications.

## 1. Introduction

Two-dimensional (2D) materials have recently garnered worldwide interest because of their photoelectric, mechanical, chemical and thermal attributes [1,2,3]. Due to their weak interlayer attraction, these 2D materials can exfoliate into monolayers or few layers. As a 2D material of great importance, graphene, with linear dispersion at the K point, has been used in diverse applications, such as electrical and thermal conductors, batteries, as a broadband optical modulator, surface plasmonics [4,5], etc. Many other 2D materials, including topological insulators (TIs) [6], transition metal dichalcogenides (TMDs) [7,8,9], antimonene [10], bismuthene [11], perovskites [12], MXenes [13,14] and black phosphorus [15] possess complementary properties to graphene. These 2D materials have fruitful uses in optoelectronic devices [16,17,18]. On the other hand, Q-switched fiber lasers have found wide applications in industry, military, communications and medicine in recent decades. Generally, the passive Q-switching laser pulse feature depends on the SA quality [19,20,21]. In the last few years, the 2D nanomaterials mentioned above have been widely used as SAs in pulsed lasers, which possess the advantages of wideband nonlinear optical response, high third-order nonlinear susceptibility, high carrier mobility and powerful interaction between light and materials [22,23,24].

In particular, TMDs have received much attention for pulsed lasers applications. TMDs have the formula MX_2_, where M is a transition metal element from the IVB group (Ti or Zr), group VB (V or Nb), group VIB (Mo or W) or group VIIB (Re), and X is a chalcogen (S, Se or Te) [25,26]. Based on group VIB (Mo and W) TMDs, the corresponding SA devices have been successfully used for different central wavelengths pulsed lasers [27,28]. Very recently, ReS_2_ also has been used for pulse generation in EDF lasers [29,30]. In contrast to the mentioned group, VIB or VIIB TMDs, with ZrS_2_ as a group IVB TMD, possess some unique photoelectric characteristics [31]. For example, ZrS_2_ 2D nanomaterials have a high electronic mobility, i.e., 1200 cm^2^/Vs, which is 3 times larger than that of the widely investigated MoS_2_ (340 cm^2^/Vs). Compared with the bandgap of WS_2_ (2.1 eV) and MoS_2_ (1.9 eV), ZrS_2_ has a smaller direct band gap, i.e., 1.4 eV, making ZrS_2_ more applicable in the near-infrared wavelength regime. Consequently, ZrS_2_ is regarded as a splendid nanomaterial which possesses remarkable optical and physical characteristics and will advance the development of optoelectronic devices. ZrS_2_-based photodetectors have been reported [32]. Therefore, it is of great significance to apply ZrS_2_ nanomaterials to the area of nonlinear optics.

In this work, we demonstrate the nonlinear optical saturable absorption properties of ZrS_2_. The ZrS_2_-based SA was prepared by depositing a ZrS_2_ nanostructured film on the side surface of a D-shaped fiber, which possessed a modulation depth (*ΔT*) of 3.3%, a saturable intensity (*I*_sat_) of 13.26 MW/cm^2^ and nonsaturable loss (*α*_ns_) of 17%, respectively. By using a ZrS_2_-based SA, passive dual-wavelength Q-switched operation is established in an EDF laser. The maximum single pulse energy is 53.0 nJ, and the shortest pulse width is 2.3 μs. As far as we know, this is the first presentation of a dual-wavelength Q-switched fiber laser with ZrS_2_. Our work demonstrates that the ZrS_2_ nonlinear optical material has bright prospects for pulsed laser applications.

## 2. Materials and Methods

### 2.1. ZrS_2_-Based SA Fabrication

Weak bonding between the ZrS_2_ layers makes it possible to separate ZrS_2_ bulk into a layered structure using liquid phase exfoliation technology [33]. Five milligrams of ZrS_2_ powder was dispersed into 10 mL isopropyl alcohol (IPA) solvent. Then, the dispersion underwent a 5 h ultrasonic process. Finally, the stable ZrS_2_ nanostructured film dispersion was obtained by further centrifuging at 6000 r/min for 30 min. Figure 1a shows a photograph of the ZrS_2_ dispersion, which exhibits a light yellow color and homogeneous features. A scanning electron microscopy (SEM) image is displayed in Figure 1b, indicating that the ZrS_2_ has layered structure. Raman spectrum is a common and useful technique for the characterization of 2D materials [34]. Figure 2 shows the Raman spectrum results of ZrS_2_. The employed wavelength is 532 nm. The in-plane mode (E_g_) and out of plane mode (A_1g_) can be clearly observed in Figure 2. The A_1g_ peak is located at 334 cm^−1^ and the E_g_ peak at 245 cm^−1^. As recently reported, the Raman signal depends on the thickness of the ZrS_2_. The A_1g_ and E_g_ peaks enhance with increasing the number of layers. Attributed to non-harmonic effects, the A_1g_ peak would show broadening trend [35,36]. In order to enhance the interaction between the laser and ZrS_2_ nanomaterials, we use the indirect evanescent field coupling method. Evanescent field coupling-based SA is formed by depositing the a ZrS_2_ nanostructured film onto the side surface of a D-shaped fiber. The D-shaped surface is 5 μm away from the fiber core, and the length of the D-shaped area is 5 mm. The ZrS_2_-D-shaped fiber SA is obtained by depositing the ZrS_2_ nanostructured film onto the side surface when the laser passes through the fiber.

### 2.2. Q-Switching Fiber Laser

Figure 3 shows a diagram of EDF laser with ZrS_2_-based SA. The laser cavity shows the ring-shaped structure. An EDF of 40 cm in length was used as the gain medium, whose absorption coefficient at 976 nm was 110 dB/m. A laser diode (LD) with a maximum power of 650 mW is adopted as pump source. A wavelength division multiplexer (WDM) is inserted into the laser cavity between the LD and EDF. In order to guarantee the laser pulse transmits unidirectionally, a polarization independent isolator (PI-ISO) was used. The polarization state in the laser cavity was adjusted using a squeezed polarization controller (PC). The ZrS_2_-D-shaped fiber SA was spliced into the laser cavity for pulse generation. A fiber optical coupler (OC) was used with a 10% laser power output. The radiation spectra are measured by an optical spectrum analyzer (Yokogawa AQ6370D, Tokyo, Japan) with a minimum resolution of 0.02 nm. The optical pulse performance was monitored using a 1 GHz digital oscilloscope (Rohde & Schwarz RTO1014, Munich, Germany) combined with a 5 GHz InGaAs photodetector (Thorlabs DET08CFC, Newton, NJ, USA). The average output power was measured using a digital power meter (JDSU OLP-85, San Jose, CA, USA).

## 3. Results

### 3.1. Nonlinear Optical Characteristics of ZrS_2_-Based SA

The commonly-used balanced twin-detector measurement scheme was adopted to research the nonlinear optical characteristics of a ZrS_2_-based SA [37,38]. A mode-locked EDF laser was used as the source, which featured a central wavelength of 1550 nm, a pulse width of 500 fs and a repetition rate of 23 MHz. Figure 4 presents the saturable absorption curve with an increasing peak power intensity. As the intensity grows, the nonlinear optical transmission increases to 83% and remains saturated. The modulation depth (*ΔT*), saturable intensity (*I*_sat_) and nonsaturable loss (*α*_ns_) are estimated to be around 3.3%, 13.26 MW/cm^2^ and 17%, respectively.

### 3.2. ZrS_2_ Q-Switched Fiber Laser

The Q-switching laser pulses featured at different pump power levels were investigated during the experiment, as shown in Figure 5. In the 90–479 mW pump range, the fiber laser operates in Q-switching state. The laser pulse trains show no pulse modulation with increasing pump power, indicating that the fiber laser works in a stable Q-switching state. The Q-switching radiation spectra are also measured. The single wavelength spectra at 90 and 110 mW are depicted in Figure 6a. It was observed that the single wavelength spectra tended to widen with increasing pump power from 90 mW to 110 mW. During the experiments, a dual-wavelength spectrum appears gradually with increasing pump power. Figure 6b depicts the dual-wavelength spectrum at a pump power of 406 mW, which shows two wavelength peaks at 1559 and 1563 nm. In order to explore the dual-wavelength Q-switching operation more substantially, the dual-wavelength evolutionary process was recorded. The corresponding radiation spectra are measured at pump powers of 144 mW, 180 mW, 217 mW, 236 mW, 406 mW, 439 mW and 479 mW, as displayed in Figure 7. With increasing pump power, the 1563 nm wavelength peak gradually enhanced. At a pump power of 236 mW, the 1559 nm and 1563 nm wavelength peaks had the same intensities, i.e., −18.9 dB. The dual-wavelength Q-switching operation can be maintained with the pump power increasing to 479 mW.

## 4. Discussion

Figure 8a shows the repetition rate increase and pulse duration decrease with a pump power increase from 90 to 479 mW, which is consistent with the typical properties of Q-switched pulses [39,40]. It was observed that the repetition rate was tuned from 16.39 kHz to 76.92 kHz. In the pump power range of 90–217 mW, there is a sharp decline in the pulse width, from 10.0 μs to 3.1 μs. In contrast, when the pump power was tuned to within the range of 256–479 mW, the pulse duration remained almost unchanged. The shortest pulse width, i.e., 2.3 μs, was obtained at a pump power of 479 mW. Figure 8b shows the dependence of the average output power and single pulse energy on the pump power. It is noted that the average output power increases linearly with pump power. When the pump power reaches 479 mW, the maximum average output power is 4.08 mW. The corresponding maximum single pulse energy is 53.0 nJ. By continuing to increase the pump power slowly, Q-switched laser pulses could not be observed. When reducing the pump power to 479 mW, the optical pulse trains appeared again. The effect of ZrS_2_-based SA on Q-switching pulses generation can be explained by the basic energy-band principle with a two level saturable absorption model [41,42]. Under an incident light with photon energies higher than the bandgap energy, photon-induced electrons are excited from the valance band to the conduction band. After photo-excitation, hot electrons in the conduction band quickly thermalize to Fermi-Dirac distribution. Subsequently, the thermalized electrons further cool down and electron-hole recombination dominates. This process may account for the linear absorption under weak light excitation. However, under strong laser excitation, most electrons in the valance band are excited to the conduction band in a very short time, which can cause the states in conduction band to be filled, thereby blocking further absorption. As a result, low-intensity lasers experience large lossed, while high-intensity lasers experience small losses. Dual-wavelength operation was achieved in this work based on the combination of the high nonlinear effect of the ZrS_2_-based SA and the spectral filtering effect [43,44,45]. By using the D-shaped fiber, the interaction between the laser and ZrS_2_ materials increases. Therefore, the prepared ZrS_2_-based SA possesses high optical nonlinearity. In addition, the D-shaped fiber has a slightly residual polarization asymmetry effect, so it can be used as a birefringence device, which induces a spectral filtering effect [46].

The comparison experimental results of the dual-wavelength Q-switched fiber lasers with 2D materials as SAs are shown in Table 1. With ZrS_2_ as the SA, a pulse width of 2.3 μs and single pulse energy of 53.0 nJ are obtained. With graphene as the SA, a single pulse energy of 70.2 nJ is presented, while the shortest pulse width is 8.2 μs. Based on Bi_2_Se_3_, the shortest pulse width, i.e., 1.16 μs, is obtained; the corresponding pulse energy is just 2.09 nJ. The average output power of 4.08 mW is larger than that of other results. In addition, our dual-wavelength Q-switching fiber laser works in a broader pump range. Overall, our experimental results are better. The better experimental results benefit from the excellent properties of the ZrS_2_-based SA. In this work, we used the indirect evanescent field coupling method to enhance the interaction between the laser and the ZrS_2_ nanomaterials. Based on this scheme, the prepared SA can enhance the nonlinear modulation effect, which is helpful for the generation of dual-wavelength Q-switching pulses in fiber lasers. The indirect evanescent field coupling method also improves the damage threshold of ZrS_2_-based SA; as such, it can endure higher laser power. That is why the dual-wavelength Q-switched fiber can work in broader pump range. In addition, the D-shaped fiber deposited with ZrS_2_ can be easily incorporated into the all-fiber laser cavity without splice loss. The inset loss can be deduced in a very small value. The pulse width could be further shortened by reducing the laser cavity length or enhancing the modulation depth of the ZrS_2_-based SA. The pulse energy could be scaled up by simply enhancing the output coupling ratio. We sought to verify whether the Q-switching operation is attributable purely to the saturable absorption of the ZrS_2_-based SA. To this end, the ZrS_2_-based SA was removed from the laser oscillator. In this kind of situation, we could not observe the Q-switching operation despite of rotating the PC and adjusting the pump power. The comparative results show that Q-switching operation is indeed attributable to the saturable absorption of the ZrS_2_-based SA.

## 5. Conclusions

In conclusion, the nonlinear saturable absorption of ZrS_2_-based SA was studied. The SA showed a modulation depth of 3.3%, a saturable intensity of 13.26 MW/cm^2^, and nonsaturable loss of 17%. Using the ZrS_2_-based SA in EDF laser, the fiber laser can work in a stable dual-wavelength Q-switching state. The output power increases linearly to 4.08 mW, corresponding to a single pulse energy of 53.0 nJ. The pulse duration can be tuned from 10.0 μs to 2.3 μs. It is the first demonstration of ZrS_2_ 2D materials exhibiting nonlinear optical absorption. The experimental results show that ZrS_2_ can be developed into an excellent candidate of nonlinear optical devices.

## Figures and Tables

**Figure 1 nanomaterials-09-00315-f001:**
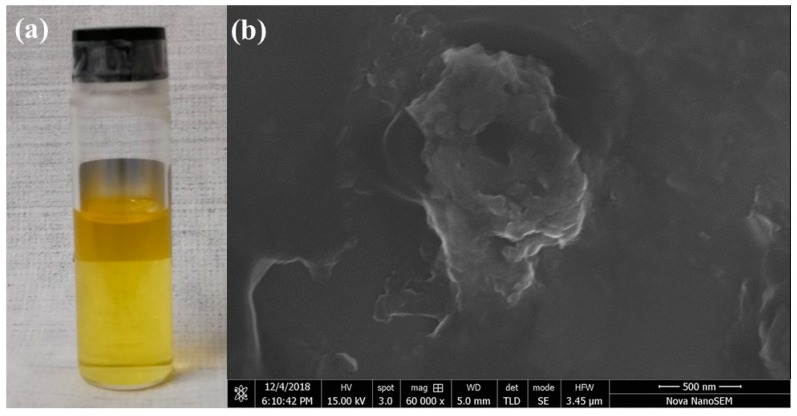
(**a**) Photograph ZrS_2_ dispersion; (**b**) SEM image of ZrS_2_ dispersion.

**Figure 2 nanomaterials-09-00315-f002:**
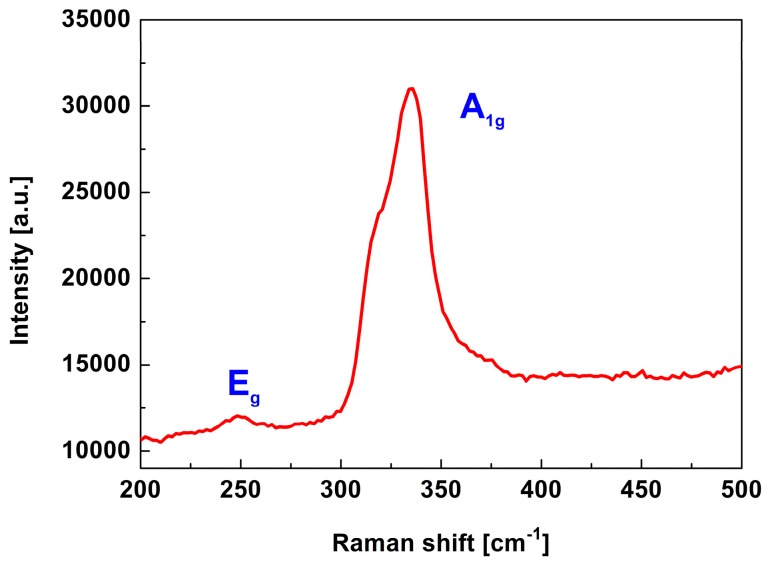
Raman spectrum of ZrS_2_ nanostructured film.

**Figure 3 nanomaterials-09-00315-f003:**
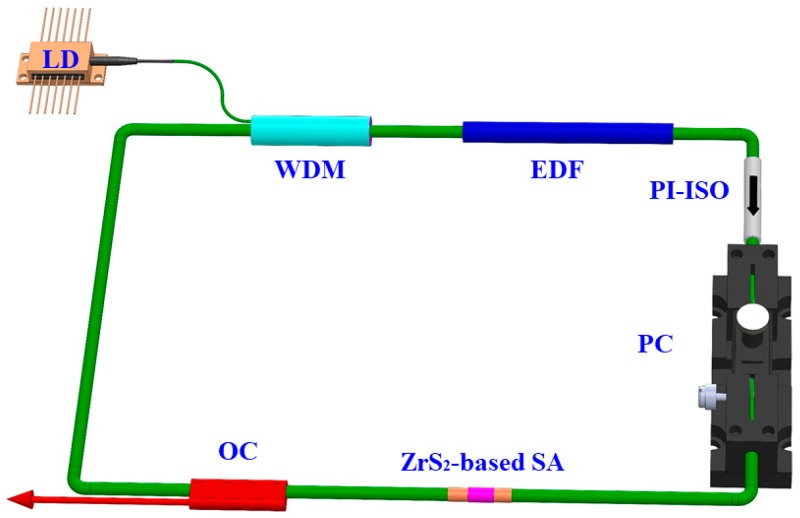
Setup of the EDF laser with ZrS_2_-based SA.

**Figure 4 nanomaterials-09-00315-f004:**
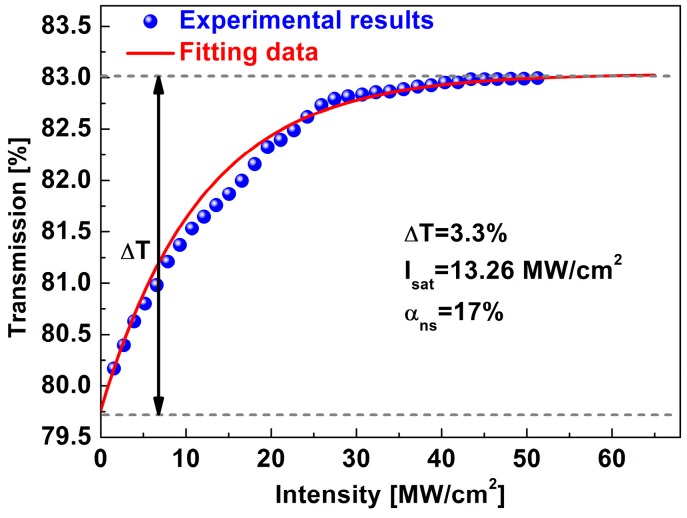
Nonlinear saturable absorption of ZrS_2_-based SA.

**Figure 5 nanomaterials-09-00315-f005:**
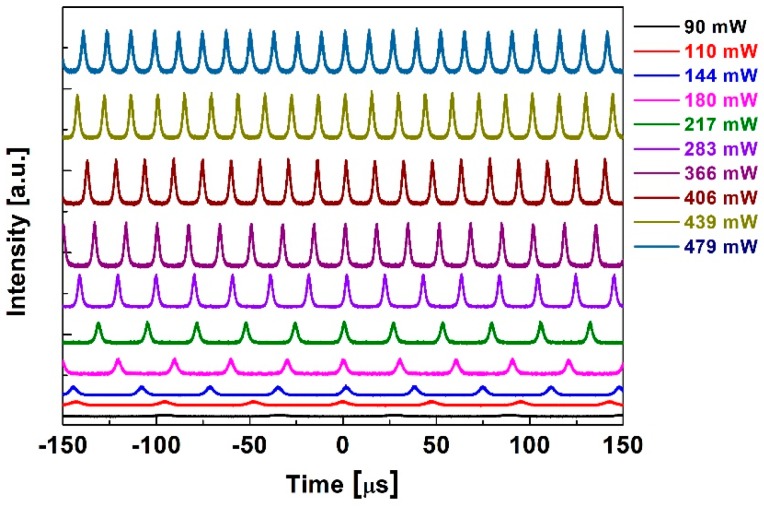
Pulse trains at different pump power.

**Figure 6 nanomaterials-09-00315-f006:**
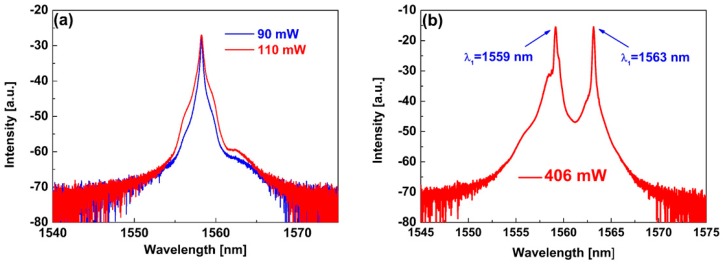
(**a**) Single wavelength radiation spectrum; (**b**) Dual-wavelength radiation spectrum.

**Figure 7 nanomaterials-09-00315-f007:**
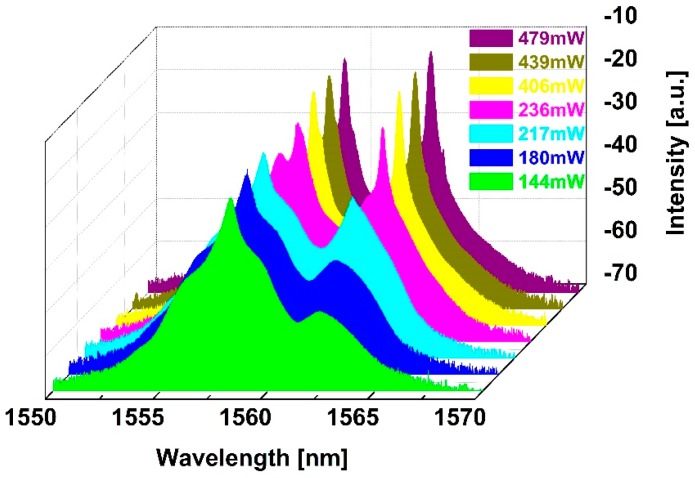
Dual-wavelength evolutionary process.

**Figure 8 nanomaterials-09-00315-f008:**
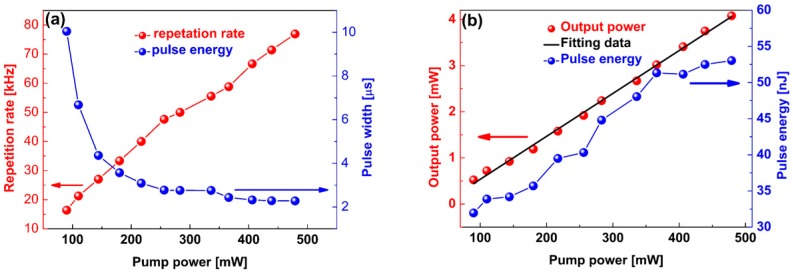
(**a**) Repetition rate and pulse duration versus different pump power; (**b**) Output power and pulse energy versus different pump power.

**Table 1 nanomaterials-09-00315-t001:** Performance comparison of dual-wavelength Q-switched fiber lasers with 2D materials.

SA	Pulse Width (μs)	Pulse Energy (nJ)	Output Power (mW)	Pump Range (mW)	Ref.
Graphene	3.7	16.7	1.1	6.5–82.8	[47]
Graphene	8.2	70.2	3.08	123.5–402.6	[43]
Graphene oxide	--	8.98	0.167	90–206	[48]
Bi_2_Se_3_	8.46	0.65	0.038	135.5–195.3	[49]
Black phosphorus	1.16	2.09	0.12	115.2–188	[50]
ZrS_2_	2.3	53.0	4.08	144–479	This work

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
