# Peer review of "Optical Nonlinearity of ZrS2 and Applications in Fiber Laser"

_nanomaterials, 2019, doi:10.3390/nano9030315_

Reviewer 1 Report

The strength of the work is that authors explored another TMD material as a saturable absorber in Q-switched lasers that is showing a better potential (compared to other attempts) to deliver a superior performance for Q-switched lasers. A weakness is coming from the fact that authors seems satisfied with their initial results and did not want to go in-depth and explore laser characteristics with the ZrS2 absorber in different optical designs as concerned the laser as well as they limited their discussions to mostly outline what they had experimentally observed.

Authors should provide a better and more comprehensive discussions on the obtained results. In particular:

1) Dual wavelength operation. Any analysis of the regime from the standpoint of gain saturation?

2) Authors claim that the laser performance (in two-wavelength regime) is better than what has been previously reported. What do they mean by better performance (extracted energy, peak power)? 

3) Why ZrS2-based Q-switching mechanism delivers the better performance? Which characteristic of the material ensures the better performance?

4) How do they make sure that the SA is a single layer sheet?

5) What happens if the material is moved within the laser cavity so that the  beam interacts with few-or-many-layers part of ZrS2?

6) Any ideas of now the Q-switched pulse can be shortened?

7) What are the ways of scaling the extracted laser energy (above 100 nJ level)?

8) Significant editing is required to bring the manuscript to a standard that is required by the journal and helps a reader to better understand it. Even sentences within the abstract are written with grammar mistakes and can make readers to get confused(e.g. second sentence, (usage of "gap") or  In this work, we investigate the nonlinear optical property of ZrS2. " ((which one?); "The experimental results indicates (?) that the ..", etc.)

Author Response

Manuscript ID: nanomaterials-442231

Title: Optical Nonlinearity of ZrS2 and Applications in fiber laser

 Dear Editors and Reviewers,

Attached please find our revised version of nanomaterials-442231, in which we have considered all the critiques and questions mentioned in the Reviewers’ reports and made corresponding modifications.

We accept the advice of the reviewers and revise our manuscript point-by-point. All the changes have been highlighted in red.

Yours sincerely,

Lu Li

Response to Reviewers

The strength of the work is that authors explored another TMD material as a saturable absorber in Q-switched lasers that is showing a better potential (compared to other attempts) to deliver a superior performance for Q-switched lasers. A weakness is coming from the fact that authors seems satisfied with their initial results and did not want to go in-depth and explore laser characteristics with the ZrS2 absorber in different optical designs as concerned the laser as well as they limited their discussions to mostly outline what they had experimentally observed.

Authors should provide a better and more comprehensive discussions on the obtained results. In particular:

1) Dual wavelength operation. Any analysis of the regime from the standpoint of gain saturation?

Reply: Great thanks for reviewer’s suggestion. The dual-wavelength operation is achieved in this work based on the combination of the high nonlinear effect of the ZrS2-based SA and the spectral filtering effect. ZrS2 materials has large optical nonlinearity. By using the D-shaped fiber, it increases the interaction between laser and ZrS2 materials. Therefore, the prepared ZrS2-based SA possesses high optical nonlinearity. In addition, the D-shaped fiber has the slightly residual polarization asymmetry effect, so it can be used as the birefringence device, which induce the spectral filtering effect. We have added the discussion in revised manuscript.

2) Authors claim that the laser performance (in two-wavelength regime) is better than what has been previously reported. What do they mean by better performance (extracted energy, peak power)?

Reply: Thanks for reviewer’s question. In table 1, we compare the experimental results (pulse width, pulse energy, output power, pump range) of dual-wavelength Q-switched fiber lasers with 2D materials. It is noted that the dual-wavelength Q-switched fiber laser with graphene as SA emits the single pulse energy of 70.2 nJ and shortest pulse width of 8.2 μs. Based on Bi2Se3, the shortest pulse width of 1.16 μs and single pulse energy of 2.09 nJ are obtained. In our work, with ZrS2 as SA, the pulse width of 2.28 μs and single pulse energy of 53.04 nJ are obtained. For the output power comparison, the average output power of 4.08 mW is larger than that of other results. In addition, our dual-wavelength Q-switching fiber laser works in broader pump range (144-479 mW). Compared comprehensively, our experimental results are better.

3) Why ZrS2-based Q-switching mechanism delivers the better performance? Which characteristic of the material ensures the better performance?

Reply: Thanks for reviewer’s question. The third-order nonlinear optical property is helpful for the generation of dual-wavelength Q-switching pulses in fiber laser. In this work, we use the indirect evanescent field coupling method to enhance the interaction between laser and ZrS2 nanomaterials. Based on this scheme, the prepared SA can enhance the nonlinear modulation effect. In addition, the D-shaped fiber deposited with ZrS2 can be easily incorporated in the all-fiber laser cavity without splice loss. As most of the power intensity distributes in fiber core, only a small part of optical power leaks from the fiber, so the inset loss can be deduced in a very small value. Based on these factors, our dual-wavelength Q-switched fiber laser delivers the better performance. We have added the discussion in revised manuscript.

4) How do they make sure that the SA is a single layer sheet?

Reply: Thanks for reviewer’s question. In this work, liquid phase exfoliation technology is used to prepare the ZrS2 nanosheets dispersion. This method is a simple and effective technique to prepare high quality few-layer structure. So the ZrS2 dispersion we used has few-layer structure. In future work, we would try to use single layer structure ZrS2 for pulse generation. Chemical vapor deposition method would be an ideal way to prepare the single layer structure.

5) What happens if the material is moved within the laser cavity so that the beam interacts with few-or-many-layers part of ZrS2?

Reply: Thanks for reviewer’s question. In this work, D-shaped fiber deposited with ZrS2 can be used as SA, which is an indispensable component for pulse generation. Our laser cavity structure including position of optical components and length of EDF is a proven design for the dual-wavelength Q-switching pulses generation. When the ZrS2-based SA is moved within laser cavity, the fiber laser only can emit common single wavelength Q-switching pulses, but not the dual-wavelength Q-switching pulses.

6) Any ideas of now the Q-switched pulse can be shortened?

Reply: Thanks for reviewer’s question. The pulse width could be further shortened by reducing the laser cavity length or enhancing the modulation depth of the prepared SA. We have added the discussion in revised manuscript.

7) What are the ways of scaling the extracted laser energy (above 100 nJ level)?

Reply: Thanks for reviewer’s question. The pulse energy could be scaled up simply by using 30% or 50% optical coupler, which would be expected to obtain 100 nJ single pulse energy output. We have added the discussion in revised manuscript.

8) Significant editing is required to bring the manuscript to a standard that is required by the journal and helps a reader to better understand it. Even sentences within the abstract are written with grammar mistakes and can make readers to get confused (e.g. second sentence, (usage of "gap") or "In this work, we investigate the nonlinear optical property of ZrS2." ((which one?); "The experimental results indicates (?) that the ..", etc.)

Reply: Great thanks for reviewer’s suggestion. We have improved the quality of the language throughout the manuscript. For the usage of “gap”, we revised the sentence as “While for the group IVB TMDs, the applications in this field remain unexplored.” The nonlinear optical properties mean the modulation depth, saturable intensity, and nonsaturable loss. In this work, we use the ZrS2-based SA for pulse generation in Er-doped fiber laser. So the experimental results mean the shortest pulse width of 2.28 μs, pulse energy of 53.04 nJ and output power of 4.08 mW. If the language are still don't meet the journals standards, please point out again.

Reviewer 2 Report

In their manuscript, Lu Li and coworkers demonstrate that ZrS2 can be used as a saturable absorber (SA) for pulsed fiber lasers. The authors show that the ZrS2-based SA has modulation depth of 2.9%, saturable intensity of 12.3 MW/cm^2 and non saturable loss of 16.8%. Using the ZrS2-based SA in Erbium-dope fiber laser, the fiber laser can work with the maximum single pulse energy of 53.04nJ and the shortest pulse width of 2.28 \mu s. The result is interesting however, if compared to other publications in journals with the similar impact factors such as Ref.[31] -[35], the results and the data analysis in the paper are not good enough to merit a publication in Nanomaterials.

1. The authors just stopped at describing results, and didn’t analyze much the results. 

2. The manuscript doesn’t include enough important references, for example the discussion on Raman spectrum is short without any reference. Except the introduction, there is no reference in other part of the manuscript.   

3. The maximum intensity in Fig.4 is still far from the saturation. The presentation of the figure is not as good as ones in other papers, for example Ref. [31] -[35].

4. The author didn’t include all materials in Table 1, for example, MoS2 can have shorter pulse width, greater maximum pulse energy and output power as listed in table 1 of Rongfei Wei et. al. Nanoscale, 8, 7704, 2016.

Author Response

Manuscript ID: nanomaterials-442231

Title: Optical Nonlinearity of ZrS2 and Applications in fiber laser

Dear Editors and Reviewers,

Attached please find our revised version of nanomaterials-442231, in which we have considered all the critiques and questions mentioned in the Reviewers’ reports and made corresponding modifications.

We accept the advice of the reviewers and revise our manuscript point-by-point. All the changes have been highlighted in red.

Yours sincerely,

Lu Li

Reviewer 3 Report

The authors’ work is devoted to the using material as a saturable absorber for the laser radiation generation in the passive mode. The problem is not new, as evidenced by a large number of works, including the authors of this study. As a novelty in this work, it could be noted the use of ZrS2, a material that has been studied since the mid-20th century. Unfortunately, the work contains an unacceptable amount of unsubstantiated conclusions, terminological errors and typographical errors in the text, therefore it cannot be recommended for publishing as is. The physical meaning is often not clear, in particular:

1. What are the reproducibility of pulse energy and the duration of laser pulses? Why the values are presented with such accuracy, for example, 53.04 nJ or 10.04 μs?

2. The authors unreasonably call the obtained ZrS2 deposits as nanosheets. This is not visible in Figure 1b.

3. There is no description of either the experimental setup or the measurement method of the “optical spectra”. Therefore, the physical meaning of this term is not clear and it is not possible to assess the reliability of the results presented in Figures 6 and 7.

4. The effect of the saturable absorber on q-switch generation is not shown.

5. The figure is not given to the journal standards.

Work requires major revision before resubmission.

Author Response

Manuscript ID: nanomaterials-442231

Title: Optical Nonlinearity of ZrS2 and Applications in fiber laser

Dear Editors and Reviewers,

Attached please find our revised version of nanomaterials-442231, in which we have considered all the critiques and questions mentioned in the Reviewers’ reports and made corresponding modifications.

We accept the advice of the reviewers and revise our manuscript point-by-point. All the changes have been highlighted in red.

Yours sincerely,

Lu Li

Round  2

Reviewer 1 Report

The manuscript has been revised, especially its 'discussion' section. I recommend publishing it in Nanomaterials.

Author Response

Thanks for the reviewer.

Reviewer 2 Report

The manuscript has been improved and is suitable for publication.

Author Response

Thanks for the reviewer.

Reviewer 3 Report

1. The authors present the results of experimental values of the laser pulses energy and duration with an accuracy greater than 0.01%. Such a presentation has no physical meaning. I recommend to provide data with an accuracy of no more than a few percent, which is typical for optical measurements.

2. The SEM image shows a monolithic sample consisting of nanoscale fragments. Nevertheless, it is impossible to estimate the thickness of the fragments in the image. Evaluation of the thickness of flakes by Raman spectrum is carried out only as a limitation from below. Therefore, the size of 4-5 nm is unproven. I recommend either to demonstrate experimental evidence for the presence of nanosheets, or use the term “nanostructured film”.

3. The “optical spectrum” is a broad concept. I recommend using more specific terms "radiation spectrum", "laser generation spectrum", etc.

4. It is necessary to correct English in some places. For example, a graph with one Raman spectrum should be signed as a “spectrum” and not a “spectra” and so on.

Author Response

Manuscript ID: nanomaterials-442231

Title: Optical Nonlinearity of ZrS2 and Applications in fiber laser

Dear Reviewer,

Attached please find our revised version of nanomaterials-442231, in which we have considered all the critiques and questions mentioned in the Reviewer’ reports and made corresponding modifications.

We accept the advice of the reviewer and revise our manuscript point-by-point. All the changes have been highlighted in red.

Yours sincerely,

Lu Li

Response to Reviewers

1. The authors present the results of experimental values of the laser pulses energy and duration with an accuracy greater than 0.01%. Such a presentation has no physical meaning. I recommend to provide data with an accuracy of no more than a few percent, which is typical for optical measurements.

Reply: Thanks for reviewer’s suggestion. We have modified the data accuracy and marked in red in revised manuscript.

2. The SEM image shows a monolithic sample consisting of nanoscale fragments. Nevertheless, it is impossible to estimate the thickness of the fragments in the image. Evaluation of the thickness of flakes by Raman spectrum is carried out only as a limitation from below. Therefore, the size of 4-5 nm is unproven. I recommend either to demonstrate experimental evidence for the presence of nanosheets, or use the term “nanostructured film”.

Reply: Thanks for reviewer’s suggestion. We use the “nanostructured film” term and marked in red in revised manuscript.

3. The “optical spectrum” is a broad concept. I recommend using more specific terms "radiation spectrum", "laser generation spectrum", etc.

Reply: Thanks for reviewer’s suggestion. We have replaced “optical spectrum” with “radiation spectrum” and marked in red in revised manuscript.

4. It is necessary to correct English in some places. For example, a graph with one Raman spectrum should be signed as a “spectrum” and not a “spectra” and so on.

Reply: Thanks for reviewer’s suggestion. We have corrected as the “Raman spectrum” and marked in red in revised manuscript.
